# Biological Activity of High-Purity β-1,3-1,6-Glucan Derived from the Black Yeast *Aureobasidium pullulans*: A Literature Review

**DOI:** 10.3390/nu13010242

**Published:** 2021-01-16

**Authors:** Toshio Suzuki, Kisato Kusano, Nobuhiro Kondo, Kouji Nishikawa, Takao Kuge, Naohito Ohno

**Affiliations:** 1Research and Development Laboratories, Fujicco, Co., Ltd., 6-13-4 Minatojima-Nakamachi, Chuo-ku, Kobe, Hyogo 650-8558, Japan; 2Aureo Co., Ltd., 54-1, Kazusa Koito, Kimitsu-shi, Chiba 292-1149, Japan; kusano-kisato@aureo.co.jp; 3Research and Development Division, Itochu Sugar Co., Ltd., 3, Tamatsuura, Hekinan, Aichi 447-8506, Japan; nobuhiro.kondo@itochu-sugar.co.jp; 4Innovation Center, Osaka Soda Co., Ltd., 9, Otakasu-cho, Amagasaki, Hyogo 660-0842, Japan; knishika@osaka-soda.co.jp; 5Life Science Materials Laboratory, ADEKA Corporation., 7-2-34, Higashi-Ogu, Arakawa-ku, Tokyo 116-8553, Japan; 6Tokyo University of Pharmacy and Life Sciences, 1432-1, Horinouchi, Hachioji, Tokyo 192-0392, Japan; ohnonao@toyaku.ac.jp

**Keywords:** *Autreobasidium pullulans*, β-1,3-1,6-glucan, physiological function

## Abstract

The black yeast *Aureobasidium pullulans* produces abundant soluble β-1,3-1,6-glucan—a functional food ingredient with known health benefits. For use as a food material, soluble β-1,3-1,6-glucan is produced via fermentation using sucrose as the carbon source. Various functionalities of β-1,3-1,6-glucan have been reported, including its immunomodulatory effect, particularly in the intestine. It also exhibits antitumor and antimetastatic effects, alleviates influenza and food allergies, and relieves stress. Moreover, it reduces the risk of lifestyle-related diseases by protecting the intestinal mucosa, reducing fat, lowering postprandial blood glucose, promoting bone health, and healing gastric ulcers. Furthermore, it induces heat shock protein 70. Clinical studies have reported the antiallergic and triglyceride-reducing effects of β-1,3-1,6-glucan, which are indicators of improvement in lifestyle-related diseases. The primary and higher-order structures of β-1,3-1,6-glucan have been elucidated. Specifically, it comprises a single highly-branched glucose residue with the β-1,6 bond (70% or more) on a backbone of glucose with 1,3-β bonds. β-Glucan shows a triple helical structure, and studies on its use as a drug delivery system have been actively conducted. β-Glucan in combination with anti-inflammatory substances or fullerenes can be used to target macrophages. Based on its health functionality, β-1,3-1,6-glucan is an interesting material as both food and medicine.

## 1. Introduction

β-1,3-Glucan, a well-known functional food ingredient derived from mushrooms, has been reported to exhibit antitumor activity [1,2,3,4,5]. The Reishi mushrooms (*Ganoderma lucidum*) and other traditional medicines have been used for generations. For instance, schizophyllan derived from the Suehirotake mushroom (*Schizophyllum commune*) and lentinan derived from the Shiitake mushroom (*Lentinus edodes*) are manufactured and traded as injectable anticancer drugs. These compounds comprise β-1,3-glucan with β-1,6-branched structures. In addition to mushrooms, β-1,3-glucans are derived from ascomycetes and other fungi, such as baker’s yeast and black yeast, which also compromise β-1,6-branched structures. Furthermore, β-glucans have been derived from oat and barley, which possess repeating structures with β-1,3 and β-1,4 bonds in their unbranched primary chains. β-Glucans derived from seaweeds, such as laminaran, comprise repeating structures with β-1,3 and β-1,6 bonds (Table 1).

In particular, many studies in Europe and North America have investigated the effects of barley-derived β-glucans on lifestyle-related diseases, such as coronary heart disease and diabetes. In May 2006, the U.S. Food and Drug Administration acknowledged a health claim label stating that β-glucan “helps reduce the risk of coronary heart disease.” In October 2009, Europe also approved a health claim that β-glucan “helps maintain normal blood cholesterol.” In Japan, a Foods with Function Claim under the label of “cholesterol reduction effect and intestinal control effect” has been submitted. Recently, a study was conducted to clarify the physiological role underlying the metabolic benefits of barley β-glucan using a mouse model of high-fat diet (HFD)-induced obesity. Then, metabolic parameters, gut microbial composition, and the production of fecal short-chain fatty acids (SCFAs) were analyzed. As a result, the beneficial metabolic effects of barley β-glucan were found to be primarily due to the suppression of appetite and improvement of insulin sensitivity, which are induced by gut hormone secretion promoted via the gut microbiota, and subsequently induced the production of fecal SCFAs [6].

Furthermore, an important source of β-glucans is the cell wall of yeast, such as *Saccharomyces cerevisiae*, of which 55–60% is β-glucans. These β-glucans are built of a glucose backbone with β-1,3 linkages, from which short sidechains branch off linked by β-1,6 bonds. Approximately 30–35% of insoluble β-glucans are present in the inner layer of the yeast cell wall, 20–22% of soluble β-glucans in the middle layer, and 30% of glycoproteins in the outer layer. Therefore, β-glucans in the yeast cell wall are present in two forms such as insoluble in bases and soluble, and yeast-derived β-glucans are basically insoluble in water without sufficient purification. An important and traditional β-glucan derived from yeast is zymosan, an insoluble polymer of glucose with a nanoparticle size of 1–2 μm in diameter, demonstrating strong antibacterial properties, activation of macrophages, and induction of cytokine secretion that enhances the immune system [7]. Moreover, β-glucan from *Candida utilis* has also been introduced as a novel source of yeast β-glucan [8,9].

However, the molecular mechanisms underlying the immunostimulatory effects of β-glucan remain to be elucidated. Recently, macrophages have been reported to harbor receptors for β-1,3-1,6-glucans. Consequently, β-glucan has attracted much attention as a potent antigenic agent that stimulates innate immunity [10]. Yadomae (2000) [2] and Ohno (2000) [3] reported that if all β-1,6-branch side chains were removed from β-1,3-1,6-glucan, such as yeast β-glucan (zymosan) by a chemically oxidized reaction, little anticancer activity is observed compared with that before the reaction, suggesting that β-1,3-glucan with β-1,6-branch enhances immunoactivity.

The black yeast *Aureobasidium pullulans* produces abundant soluble β-1,3-1,6-glucan, which possesses strong immunostimulatory activity. *A. pullulans* (formerly known as *Pullularia pullulans*, *Hormonema dematioides*, or *Dematium pullulans*) is an imperfect fungus, which is common in nature and typically found growing in soil and water, as well as on weathered wood and many other plants. Recently, *A. pullulans* was taxonomically classified as an anamorph of the teleomorph *Sydowia polyspora*. In another taxonomic study, de novo genome sequencing of four *A. pullulans* strains was performed [11,12].

The recent taxonomic classification of *A. pullulans* is as follows:Kingdom: FungiPhylum: AscomycotaClass: DothideomycetesSubclass: DothideomycetidaeOrder: DothidealesFamily: DothioraceaeGenus: *Aureobasidium*Species: *A. pullulans*

In 1969, Arkadjeva first reported the presence of functional polysaccharides in *A. pullulans*. In addition [13], Han et al. (1976) used this fungus as a highly nutritious single-cell proteinaceous food produced via fermentation using ryegrass hydrolysate as the substrate [14]. β-glucan has also been produced via fermentation using soybean as the substrate. Furthermore, Anastassiadis et al. (2007) have reported gluconic acid production [15], and Bharti et al. (2013) have reported fructo-oligosaccharide production [16]. At present, functional food products containing *A. pullulans*-derived, high-purity, solubilized β-1,3-1,6-glucan produced by several Japanese companies are commercially available [5]. Moreover, the structural details and the three-dimensional structure of β-glucan have also been clarified, and studies on its use as a drug delivery system have also been conducted [4,17].

In this review, we discuss the production method, safety, and various physiological functions of β-glucan derived from the black yeast *A. pullulans* based on previous studies in MEDLINE database. We searched for articles in MEDLINE using the keywords “black yeasts,” “functionality,” and ”clinical tests.” We obtained 155 studies, including functional food cases in animals and clinical trials. Finally, a total of 47 relevant studies to biological activities, including those published by the authors of this review, were included and their results were summarized as a review of *A. pullulans*-derived β-glucan (APβG) functional food products.

## 2. Results

### 2.1. Production of APβG via Fermentation

During APβG production, food ingredients are used as the medium and sucrose as the sole carbon source for fermentation [18]. The obtained culture solution has a viscosity of several thousand cP (mPa·s). In addition, a unique manufacturing method by adjusting the metal salt concentration, pH, and temperature and reducing the viscosity of the culture solution has proven industrially useful. Accordingly, a stable fermentative production method for low-viscosity and high-purity APβG has been established.

In Japan, APβG is approved as a thickener under the label of “Existing Food Additives.” In addition, the culture solution itself is also sold as functional food, and following recovery and purification, it is also used as a high-purity β-glucan material.

#### 2.1.1. Fermentation

During fermentation, *A. pullulans* (AP) degrades sucrose as the carbon source into glucose and fructose under aerobic conditions and utilizes these sugars in that order. This is accompanied by β-glucan production. Consequently, the viscosity of the culture solution increases to several thousand cP. AP is an imperfect fungus and typically grows in the mycelial form; however, it assumes a yeast-like form at the APβG production stage—the reason it has acquired the moniker “black yeast.” AP also produces black melanin in the late phase of culture, but ascorbic acid can be used to effectively suppress melanin production [18,19].

#### 2.1.2. β-Glucan Recovery and Purification

Since the culture solution containing β-glucan has a high viscosity of several thousand cP (mPa.s) or higher, it can be directly used as a functional food material following sterilization. Upon purification, it is difficult to remove microorganisms because of its high viscosity. Therefore, various unique methods have been developed.

The black yeast culture solution is subjected to homogenization by physical stirring or a similar method, and then the yeast cells are separated through a filter press to obtain an aqueous polysaccharide solution. If necessary, APβG can be recovered as a precipitate using ethanol precipitation or a similar method and dried to obtain a powdered product of high purity [20].

β-glucan has a rigid triple helical structure under a neutral pH; however, under alkaline conditions, the structure changes to a random coil and the solution viscosity is reduced [1]. Consequently, microorganisms can be easily removed using a filter press and further dialysis steps can be performed. Finally, high-purity β-glucan can be mass produced using an alcohol (e.g., ethanol) precipitation method. Interestingly, the solubilized β-glucan thus obtained can form ultrafine particles (<200 nm) in an aqueous solution, which may be useful for stimulating intestinal immunity [18].

APβG can be precipitated by adding alum to the black yeast culture solution, following separation using a filter press to recover APβG. Thereafter, APβG is appropriately diluted with water and slurried, and then the pH is adjusted to a neutral range. Finally, the solution is subjected to hydrothermal treatment from 170 to 180 °C under 2 MPa for approximately 10 min. The purity of the solution is increased via filtration by adding diatomaceous earth or activated carbon and ultrafiltration following hydrothermal treatment; the final product is powdered using freeze drying. Of note, the hydrothermal treatment reduces molecular weight of APβG [21].

The concentration of high-purity APβG recovered by ethanol precipitation or a similar method is measured using the phenol–sulfuric acid method using glucose as a reference material; this method conforms to the one used for measuring reduced sugars. Furthermore, NMR spectroscopy can also be used for qualitative analysis. Structural analysis is performed using ^13^C-^1^H two-dimensional NMR spectroscopy. The C(1)-H signal integration values of β-1,3 and β-1,6 bonds are 4.8 and 4.5 ppm, respectively. With respect to the results, β-1,3 bonds are present in the primary chains and β-1,6 bonds in the side chain, confirming a high degree of branching. In addition, based on the analysis of enzymatic degradation by exotype β-1,3-glucanase, glucose, and gentiobiose have been confirmed as the products, and their structures have been demonstrated to harbor β-1,3 bonds in the primary chain and a single branched glucose with a β-1,6-bond showing over 70% branching. Gel chromatography using pullulan as a standard and sodium hydroxide as a developing solvent revealed that the MW of β-glucan thus produced is over 1,000,000 [18,19].

### 2.2. Structure and Bioactivity of APβG

APβG exhibits an array of physiological activities. The biological activity of β-glucan depends on its primary structure, conformation, and molecular weight. Therefore, the primary structure of APβG was clarified using NMR spectroscopy, and its physiological activity was assessed. It comprised a primary chain of β-1,3-d-glucan and side branches of β-1,6-β-d-glucopyranosyl units at every two residues as the major structure or a primary chain of β-1,3-d-glucan and a side chain of β-1,6-β-d-glucopyranosyl units every three residues as the minor structure [20]. Moreover, the higher-order conformation of APβG had a delicate triple helical structure [17].

In addition, APβG exhibits immunostimulatory effects, such as immune cell accumulation, as well as priming effects on the intestinal bacteria. According to many studies reviewed here, β-1,3-glucans with many β-1,6 glucopyranosyl branches, such as APβG, have unique structures, even though they are isolated from different organisms, including bacteria and plants, among others [5,20].

Interestingly, a unique hydrothermal process for the preparation of APβG may produce an active reagent [21]. Reprocessing APβG increased low molecular weight fractions, suppressive activities were markedly enhanced, and the resulting APβG was estimated to have a low MW of approximately 10,000. Lipopolysaccharide (LPS)-induced nitrogen oxide (NO) synthesis and tumor necrosis factor (TNF)-α production in RAW264.7 cells were suppressed by the resulting low molecular weight APβG in a dose-dependent manner. Therefore, low molecular weight fractions obtained by the hydrothermal processing of APβG may result in potential reagents that control inflammation induced by various pathogens [22].

### 2.3. Safety

The black yeast culture solution is a safe material for human consumption and has been registered as a thickener and stabilizer under the label “Existing Food Additives.” Accordingly, it is also utilized in food products in Japan. According to the “Survey and Study on Review of Safety of Existing Food Additives” by the Japan Food Chemistry Research Foundation of the Ministry of Health, Labor and Welfare Foundation (June 2004), there were no problems associated with the administration of repeated doses of this solution for 90 days or later and the results of mutagenicity test were negative, confirming the safety of the black yeast culture solution as a food ingredient [23]. On the other hand, pullulan, which is an α-1,4-1,6-glucan, produced by *A. pullulans*, is admitted as a food additive in both the United States (NO.GRN000099, generally recognized as safe (GRAS) status) and Europe (EFSA-Q-2003-138); thus, *A. pullulans* are considered safe for consumption of food use.

In addition, β-1,3-1,6-glucan is safe for use in food products and cosmetics as a high-purity commercial ingredient as we describe below. In studies of APβG-containing food products in human participants, three times the recommended dose was administered for four months or two years and no obviously abnormal findings were noted in clinical tests, indicating its safety as a food material [24]. In addition, a Japanese company has been selling the APβG-containing food product named “*Aureobasidium* cultured solution” as a dietary supplement since 1999, and the estimated amount sold exceeds several hundred tons. Of note, there have been no reports of complaints related to adverse health effects from the customers (data not shown).

### 2.4. Physiological Function of β-1,3-1,6-glucan

Various physiological functions of β-1,3-1,6-glucan have been confirmed, and its novel functions have been revealed using oral administration experiments in animal models. Clinical trials have also been conducted. Table 2 summarizes the results of such studies, and the key physiological functions of β-1,3-1,6-glucan are described below.

#### 2.4.1. Intestinal Immunostimulatory Effect

Organisms absorb nutrients in the intestine from food, assimilate them for growth and other physiological functions, and generate energy to survive. However, food contains pathogens and, therefore, an efficient immune system is required for protection against such pathogens. A prominent organ related to the immune function is the gut-associated lymphoid tissue (GALT) [68].

The gut microbes, such as lactobacilli and bifidobacteria, are important for the activation of intestinal immunity. Recent studies have revealed that polysaccharides or nucleic acid components of the cell surface, which constitutes the cellular body, are the essential ligands for the activation of innate immunity and induction of acquired immunity in the intestine. By recognizing the components of various microbial cells, innate immune activation and signaling enhance the responsiveness of acquired immunity. In other words, through exposure to food (ingredients), GALT matures, gains immune responsiveness, and accumulates and exerts its functions. In 2011, the discovery of the interactions between the innate and adaptive immunity was awarded the Nobel Prize in Physiology or Medicine.

The Peyer’s patch plays pivotal roles in the intestinal immunity, specifically the GALT immunity. At the Peyer’s patch, useful immunostimulatory components of food are taken up into the lymph node, where antigen-presenting cells, such as macrophages and dendritic cells, are present. These cells are taken in by the M cells in the Peyer’s patches, which are tens of microns in size. These antigen-presenting cells express innate immune signal receptors, such as TLRs and C-type lectins, and transmit various signals to the systemic immune system [10].

The intestinal immunostimulatory and modulatory effects of APβG have been evaluated in animal models such as mice. Cellular-level in vitro experiments using lymphocytes derived from the Peyer’s patches in mice showed that immunoglobulin (Ig)A was produced at APβG concentrations ranging from 0 to 200 μg·mL^−1^ in a dose-dependent manner, and APβG was more active than yeast-derived zymosan and bacterial inflammatory substances (e.g., LPS). Simultaneously, interleukin (IL)-5 and IL-6 production was also increased In in vivo experiments, APβG was orally administered to each mouse at a concentration of 10–20 mg·kg^−1^ body weight per day for seven7 days; on day 8, lymphocytes from the Peyer’s patches were extracted and their IgA level were measured. APβG promoted the production of IgA in the intestine, and activated lymphocytes and production of IL5 and IL6 in the Peyer’s patch, which is the immune tissue of the intestine, by triggering the intestinal immune system [25].

Furthermore, when APβG-containing food products were orally administered to six-week-old BALB/c female mice, antibody titers in the blood were elevated and the phagocytic capacity of blood macrophages was enhanced, suggesting an immunostimulatory effect via intestinal immune activation [52].

In addition, the Peyer’s patch cells were isolated, and IL-5, IL-6, and IgA levels in the medium were measured following APβG administration. The levels of both cytokines and IgA were increased, and the level of IL-6 secreted by the Peyer’s patch dendritic cells was also elevated. In another study, APβG was orally administered for two weeks, and IgA levels were measured; APβG tended to promote small intestinal IgA production. Interestingly, following treatment with the immunosuppressant cyclophosphamide (CY), mice receiving an APβG-supplemented diet showed significantly increased IgA production compared with mice receiving the control diet. Moreover, the levels of IL-6 and IgA secreted by the Peyer’s patch lymphocytes as well as of IL-6 secreted by the Peyer’s patch dendritic cells were increased. Overall, these effects of oral administration indicate the potential use of APβG as a functional food with immunomodulatory activity [46].

#### 2.4.2. Splenic Immunomodulation

Although the biological action of β-glucan depends on its structure, the effect of highly branched 1,3-β-d-glucan on cytokine production in mouse leukocytes remains poorly understood. In an in vitro study, APβG strongly induced the production of various cytokines in DBA/2 murine splenocytes. Specifically, APβG induced interferon-γ (IFN-γ), IL-12p70 (a Th1-type cytokine), and IL-17A (a Th17-type cytokine), but not IL-4, IL-10, and tumor necrosis factor-α (TNF-α) [32]. Furthermore, anti-dectin-1 neutralizing antibodies could not inhibit this APβG-induced IFN-γ production in DBA/2 murine solenocytes. This result indicates that APβG induces IFN-γ activity through signaling pathways other than those involving dectin-1, which is a major β-1,3-d-glucan receptor [37].

APβG has been reported to induce cytokine production in the presence of granulocyte–macrophage colony-stimulating factor (GM-CSF). This APβG-induced production of cytokines in DBA/2 murine splenocytes was completely blocked by anti-GM-CSF neutralizing antibodies. Moreover, the addition of GM-CSF to C57BL/6 murine splenocytes that were less responsive to APβG showed APβG-induced cytokine production. These findings suggest that GM-CSF is essential for APβG-induced cytokine production in murine splenocytes. This discovery is expected to offer novel insights into the effects of β-1,3-1,6-glucan as well as to help design and develop highly efficient β-glucan formulations [38].

#### 2.4.3. Antitumor and Antimetastatic Activities

The antitumor and antimetastatic activities of APβG have been investigated. In a murine oral administration study, 10,000 colon-26 cancer cells were transplanted into murine spleens to investigate the antitumor effects of APβG on primary tumors and its inhibitory effects on cancer metastasis to the liver; significant (*p* < 0.05) reduction in primary tumor incidence and inhibition of tumor metastasis to the liver were observed even at an APβG dose as low as 50 mg·kg^−1^. Furthermore, to clarify its mechanism of action in the intestine, the immune function of the small intestine was examined. The number of IFN-γ–positive cells was remarkably increased. Based on this result, the antitumor and antimetastatic effects of APβG were assumed to have been exerted through the enhancement of IFN-γ production from the small intestinal mucosal cells. Natural killer (NK) cell activity was also enhanced. That was the first study to demonstrate that the oral administration of high-purity β-glucan exhibits antitumor and antimetastatic activities by enhancing IFN-γ production and NK cell activity by triggering the intestinal immunity [26].

Furthermore, the mechanism underlying the antitumor effects of APβG has been explored. APβG induced the expression of TNF-related apoptosis-inducing ligand (TRAIL) in murine and human macrophage-like cells, suggesting that TRAIL expression induces tumor cell apoptosis [55].

In addition, APβG has a β-1,3-1,6 structure, similar to lentinan, which is an antitumor agent approved for clinical use in Japan. However, lentinan is intravenously injected. In three clinical cases, APβG was orally administered in combination with anticancer drugs, such as Avastin, Elplat, Levofolinate, and fluorouracil, to treat for stage III colon cancer patients, on which two sites of cologenomic cancer were transferred, and remarkable cancer elimination was reported in the all cases [60].

#### 2.4.4. Antimicrobial Activities

The efficacy of APβG was tested against *Candida* in intravenously infected mice and methicillin-resistant *Staphylococcus aureus* (MRSA) in intestinally infected mice. Mice sensitive to CY were intravenously infected with 6 × 10^4^ cells of *C. albicans*, and APβG was administered intraperitoneally at 1 mg per day for four days, which significantly prolonged survival. In CY-treated mice exhibiting a mild infection (intravenous administration of 2 × 10^4^ cells of *C. albicans*), oral administration of 2.5% APβG significantly prolonged survival and reduced renal microbial viability at 30 days of infection. In mice intestinally infected with MRSA, oral administration of APβG did not reduce fecal MRSA, although it inhibited CY-induced weight loss. Prophylactic oral administration of APβG improved the resistance of CY-treated mice to *C. albicans* infection [45].

#### 2.4.5. Effect of Alleviating Influenza Symptoms

Oral administration of APβG has been reported to protect mice from lethal influenza [caused by A/Puerto Rico/8/34 (PR8; H1N1) strain], which is an infectious disease. The survival of mice infected with a sublethal dose of the influenza A virus was significantly increased following the oral administration of APβG. Furthermore, viral titers in mouse lungs were significantly reduced following the oral administration of APβG. Therefore, APβG administration likely enhanced the expression of viral sensor molecules to exert protective effects against influenza through the inhibition of viral replication [40].

Previous studies have shown that APβG exhibited adjuvant activity and induced resistance against influenza; however, further investigation into the underlying mechanisms revealed that the intraperitoneal administration of APβG increased the serum level of IL-18 and the number of splenic IFN-γ-producing CD4-positive cells after influenza A viral infection. In addition, APβG induced IL-18 production in DC2.4 cells, a dendritic cell line, as well as in peritoneal exudate cells, including peritoneal macrophages. Thus, APβG acts as an adjuvant inducing the Th1-type response during influenza A viral infection [65].

To test its non-influenza antiviral efficacy, THP-1 macrophages were treated with APβG along with R-848, which is an anti-herpesvirus agent; the expression of TNF-α and IL-12p40 was significantly enhanced when the cells were co-stimulated with the M2 cell culture supernatants of R-848 and APβG compared to when stimulated with the M2 cell culture supernatant of R-848 alone. Furthermore, co-stimulation with R-848 and APβG significantly enhanced the phagocytosis-promoting ability of apoptotic Jurkat cells. These findings suggest that the APβG-induced activation of several distinct innate immune receptor signaling pathways enhances the overall immune response induced by R-848 [47].

As noted above, the antiviral effects of APβG may also be realized through the expression of interferon-stimulated genes (ISGs) in macrophage-like cell lines. These findings suggest that APβG stimulation effectively promotes the expression of ISGs through inducing IFNs and enhancing STAT1-mediated transcriptional activity [50].

#### 2.4.6. Effects on Improving Lifestyle-Related Diseases, Including Obesity

Oral administration of APβG modulated the development of fatty liver caused by an HFD. Increased blood cholesterol and triglyceride levels triggered by HFD intake were suppressed by the oral administration of APβG. In addition, triglyceride accumulation in the liver was significantly suppressed by the oral administration of APβG. In HFD-fed mice, the elevated serum alanine aminotransferase levels associated with hepatic injury were lowered by the oral administration of APβG. These findings indicate that the oral administration of APβG may be effective in preventing the development of nonalcoholic fatty liver disease. In this case, the concentration of oral administration was less than 1% aqueous solution. APβG led to a significant upregulation of cholesterol 7 alpha-hydroxylase (CYP7A1) and IL-6 and normalized lipid metabolism. There is a possibility that this is due to the immunity-mediated cytokine network rather than the black yeast β-glucan physically binding to bile acids. [53].

Moreover, the effect of APβG on atherosclerosis induced by HFD intake was also confirmed in apolipoprotein E-deficient mice, which is a common animal model of atherosclerosis. Atherosclerosis induced by HFD intake was significantly suppressed in APβG-treated mice compared to that in control mice. Serological analysis showed that the blood levels of oxidized low-density lipoprotein cholesterol, a well-known risk factor for atherosclerosis, were significantly reduced in APβG-treated mice. In addition, APβG reduced macrophage accumulation in the blood vessels. These data suggest that oral administration of APβG is effective in preventing the development of atherosclerosis induced by HFD intake [58].

In a clinical study, a male subject was orally administered 1.5 mg APβG for 2 months. As a result, his triglyceride levels decreased from 523 (at the start of the study) to 175 mg·dL^−1^ (after two months). Dyslipidemia is a major risk factor for the development of cardiovascular diseases, and statins are the routine drugs used to manage dyslipidemia. In this case report, the subject was a patient with dyslipidemia but without diabetes, who was treated with Rosuvastatin. Upon APβG administration, his very low-density lipoprotein level decreased from 104.6 (at the start of the study) to 35 mg·dL^−1^ after two months, whereas the high-density lipoprotein cholesterol level increased from 27 to 38 mg·dL^−1^. That was the first report on the effects of APβG on dyslipidemia not associated with diabetes. Thus, APβG supplementation along with routine medication is beneficial to treat dyslipidemia, although a large-scale prospective trial is warranted to confirm these effects [51].

#### 2.4.7. Postprandial Blood Glucose Reduction

The effects of the oral administration of APβG on increased postprandial blood glucose levels were examined. APβG was orally administered to mice at doses of 50, 100, 200, and 500 mg·kg^−1^ per day in the morning and evening for seven days. On day 8, after fasting for 4 h or more, an aqueous solution containing APβG and an aqueous glucose solution (100 mg, 0.2 mL per mouse) were orally administered. In the control group, the blood glucose levels increased up to 30 min after administration. In the APβG-treated group, the blood glucose levels decreased 30 min after administration, and the blood glucose levels at 60 min after administration were significantly decreased compared with those in the control group. In the groups administered 100 and 200 mg·kg^−1^ APβG, blood glucose level slightly increased from 15 to 30 min after glucose administration but decreased thereafter, and the blood glucose level at 60 min after administration was significantly decreased compared with the control level.

In addition to the single-dose study described above, a seven-day free-intake study using an aqueous solution containing APβG (0.1–1.0%) was also conducted. When insulin levels were measured (at 15 min after glucose administration) during the glucose tolerance test on day 8, the increase in postprandial blood glucose tended to be suppressed, and blood insulin levels were significantly decreased. Based on these results, APβG intake with a meal might be effective in controlling postprandial blood glucose and insulin levels, thereby reducing the risk of metabolic syndromes [35].

#### 2.4.8. Anti-Type I Allergic and Anti-Inflammatory Effects

The effect of APβG on type-I allergies was examined. Mice were provided an APβG-containing diet (0%, 0.25%, 0.5%, and 1.0%) ad libitum for 37 days. Food allergy was induced by intraperitoneal administration of 3 mg ovalbumin (OVA) saline on days 16 and 30 and oral administration of 15 mg OVA on day 37. As a result, the increased levels of OVA-specific IgE due to food allergy were lowered by APβG intake, and these levels were significantly decreased in mice receiving 1% APβG.

Th1-dominant response was observed due to increased IFN-γ and IL-12 production in splenic lymphocytes, and the numbers of CD8-positive, IFN-γ-positive, and IgA-positive cells were increased in the small intestinal mucosa. These results indicate that APβG produces an anti-type-I allergic effect through a mechanism suppressing OVA-specific IgE production. OVA ingestion mediates IgE production, which induces an allergic food reaction, leading to a Th1-dominant state. Moreover, APβG-mediated activation of the intestinal immune system under OVA-induced food allergies may protect against bacterial and viral invasion by increasing the numbers of CD8-positive T-cells, IgA-positive cells, and INF-γ-positive cells [28]. Furthermore, the anti-inflammatory effect of APβG may be strengthened by reducing its molecular weight [17].

APβG showed an allergy-alleviating effect through mast cells. The precise role of APβG in type-1 allergic reactions remains to be fully investigated. The inhibitory effects of low-molecular-weight β-glucan on mast cell degranulation and passive cutaneous anaphylaxis (PCA) were investigated. APβG inhibited the degranulation of both rat basophilic leukemia (RBL-2H3) and cultured mast cells (CMCs) activated by the calcium ionophore A23187 or IgE in a dose-dependent manner. Furthermore, oral administration of APβG inhibited IgE-induced PCA in mice. Specifically, a single dose of APβG (100–150 mg·kg^−1^ per mouse) significantly reduced PCA. Of note, tranilast (active ingredient of Rizaben, Kyorin Pharmaceutical) has been shown to be as effective as APβG [44]. Based on this result, a single-blind two-group parallel-monitoring study on pollinosis in humans (60 participants) was conducted. Consumption of drinks containing 150 mg APβG every day significantly reduced end-points such as sneezing and nasal discharge associated with pollinosis. The incidence of allergic diseases, such as allergic rhinitis, atopic dermatitis, asthma, and food allergies, has increased in several countries. Mast cells play critical roles in various biological processes related to allergic diseases. These cells express a high-affinity receptor for IgE on their surface, and the interaction of multivalent antigens with surface-bound IgE leads to the secretion of granule-stored mediators as well as the de novo synthesis of cytokines. These mediators and cytokines promote the development of allergic diseases [59].

The symptoms of mice with OVA-induced allergic asthma were alleviated following the administration of 125 mg·kg^−1^ APβG. In another study, the effects of APβG, derived from a UV-induced mutant of *A. pullulans*, on OVA-induced allergic asthma were examined and compared with the effects of intraperitoneally administered dexamethasone (DEXA) (3 mg·kg^−1^) in mice. Following OVA aerosol challenge, body weight, lung weight, total leukocyte and eosinophil count in the peripheral blood, total cell number, neutrophil count, and eosinophil count in bronchoalveolar lavage fluid were increased in the OVA-control group compared with values in the sham-control (non-OVA) group. Therefore, APβG produces favorable effects against OVA-induced allergic asthma. The efficacy of 125 mg·kg^−1^ APβG was similar to or slightly lower than that of 3 mg·kg^−1^ DEXA [41].

#### 2.4.9. Anti-stress and Immunomodulatory Effects

The effects of APβG administration on mice subjected to forced fasting and restraint stress were examined. The mice were orally administered APβG (25, 50, or 100 mg·kg^−1^ mice) every morning for seven consecutive days after one week of acclimatization. Forced fasting and restraint stress were induced using a 50 mL plastic tube with vents for 12 h from night to morning on days 3, 5, and 7 of administration. The restraint group, forced fasting plus restraint group (food and water were not provided plus forced restraint was performed), forced fasting group (food or water were not provided but no forced restraint was performed) were compared with the normal group (control; food and water provided ad libitum, and no forced restraint was performed).

The blood levels of corticosterone, which a stress marker, were significantly increased in the restraint and forced fasting plus restraint groups compared with those in the control group, and these levels tended to increase in the forced fasting group. In contrast, APβG treatment suppressed the increase in corticosterone levels. Specifically, the increase in blood corticosterone level was significantly suppressed in the 50 mg·kg^−1^ APβG administration group compared with that in the forced fasting plus restraint and forced fasting groups, indicating a stress-relieving effect of APβG. Therefore, the oral administration of APβG significantly alleviated restraint stress in mice.

IL-12 secretion from splenocytes was significantly reduced after forced fasting plus restraint stress. Moreover, restraint stress significantly suppressed immune function. Spleen weight was significantly lower in the forced fasting plus restraint and forced fasting groups than in the control group. In contrast, the decrease in IL-12 due to forced fasting and restraint stress was improved following APβG treatment. In particular, the IL-12 secretory volume was significantly increased in the 100 mg·kg^−1^ administration group compared with that in the forced fasting group. The efficacy of APβG in improving the immune function by restoring IL-6 secretion was also observed. The reduction in spleen weight due to restraint stress was not significantly different among the APβG-treated groups. In addition, the NK activity of splenic lymphocytes was markedly decreased in the forced fasting and forced fasting plus restraint stress groups, but APβG significantly ameliorated the decrease in NK activity caused by forced fasting and restraint stress.

Thus, repeated restraint stress increased blood corticosterone level but decreased splenic weight, splenic cytokine (IL-6 and IL-12) production, and NK activity. In contrast, the oral administration of APβG prevented the increase in blood corticosterone levels and the decrease in cytokine production and NK activity caused by restraint stress. Although restraint stress reduced immune functions, such as cytokine production and NK activity, the oral administration of APβG ameliorated this reduction in immunocompetence [29].

Along with the aforementioned anti-stress results, the authors planned and performed a clinical trial with a randomized crossover test (unpublished data). The duration of the clinical trial was two weeks and involved 28 participants (aged 22–55 years). In this trial, all participants consumed a bottle of placebo or APβG (100 mg) daily. After one and two weeks, corticosterone concentration in the saliva was measured as a stress marker. The anti-stress effect was physiologically estimated based on the corticosterone concentration after a stress test, such as the calculation and Visual Analogue Scale test (VAS) evaluation conducted on participants. The results suggested that, after two weeks, both physiological and mental stresses were reduced significantly. [5].

#### 2.4.10. Protective Effect on the Intestinal and Gastrointestinal Mucosa

The redacting effects of APβG on the side effect of oral anticancer drug cocktail tegafur/uracil (UFT) were examined in mice. In the group treated with UFT alone, diarrhea, which is one of the side effects, was observed, whereas in the group treated with both APβG (oral; 25–100 mg·kg^−1^) and UFT, diarrhea was suppressed. Further examination of the tissue condition confirmed that the oral administration of APβG markedly inhibited damage to the small intestinal mucosa [34].

In mice treated with the anticancer drug CY, the weight of the spleen and thymus was reduced and the level of the corresponding cytokines, such as IL10, was also decreased. In mice receiving APβG, this immunosuppressive effect of CY was reduced. The immunomodulatory effects of APβG were evaluated in CY-treated mice. APβG could effectively prevent CY-mediated immunosuppression, at least partially, by recruiting T-cells and TNF-α-positive cells or enhancing their activity. Therefore, APβG could effectively prevent the immunosuppressive effects of treatment regimens for diseases such as cancer, sepsis, and high-dose chemotherapy or radiotherapy [36].

The effects of APβG on gastric ulcers were also investigated. APβG was orally administered to 6–8-week-old male mice at a dose of 0, 5, 20, 50, 100, and 200 mg·kg^−1^ body weight. After 1 h, 100% ethanol was administered orally at 5 mL·kg^−1^ body weight to induce gastric injury (gastric ulcer). The stomach was then excised by laparotomy 4 h after the induction of the gastric injury and scored for hemorrhagic injury (hemorrhagic damage). The hemorrhagic injury score was calculated in 1 mm squares in the wounded areas and summed up to yield the gastric injury index. The results showed that the oral administration of 100% ethanol caused significant gastric injury and that APβG ameliorated this injury in a dose-dependent manner.

Therefore, the oral administration of APβG reduced ethanol-induced gastric injury (gastric ulcer), and this protective effect on the intestinal mucosa was mediated through the induction of heat shock protein 70 (HSP70) and increased production of mucin [39].

#### 2.4.11. Effects on the Microbiome

The effects of APβG on the microbiome have also been demonstrated in domestic animals. APβG was orally administered for three months to Holstein cows fed raw milk with a somatic cell count of 3 × 10^5^ mL^−1^ or less in milk, and the concentrations of milk and blood cytokines were analyzed. The number of somatic cells in milk did not change significantly but the concentration of solid non-fat in milk tended to increase following the oral administration of APβG. Analysis of blood cytokine levels using enzyme-linked immunosorbent assay revealed that TNF-α and IL-6 expression was slightly lower in the APβG-administered cows than in the control cows after two months of oral administration. Moreover, the expression of IL-8 tended to be moderately higher in the APβG-administered cows than in the control cows after three months of oral administration. APβG was orally administered to peripartum Japanese Black beef cows and their newborn calves, and the intestinal flora of the calves was analyzed using terminal restriction fragment length polymorphism. The intestinal flora was altered following the oral administration of APβG. For instance, the population of the genus *Prevotella* (317 bp operational taxonomic units) tended to increase in the APβG-administered calves compared with that in the control calves. Therefore, the oral administration of APβG may affect the blood cytokine levels in Holstein cows and the intestinal flora in Japanese Black calves [42].

The effects of APβG produced via fermentation using soybean as the substrate on the intestinal morphology of broiler chickens were examined. Improved gut microbiota (increased ratio of lactobacilli and *Clostridium*
*perfringens*) and weight gain were observed. Soybean hulls (SBH) are a by-product of soybean processing for oil and meal production. *Pleurotus eryngii* stalk residues (PESR) are a by-product of the edible portion of the fruiting body enriched in bioactive metabolites. The effects of APβG produced via the co-fermentation of PESR and SBH on the performance and intestinal morphology of broiler chickens were evaluated [56]. A total of 400 broiler chicken (Ross 308) were randomly assigned into four groups receiving the basal diet (control) or the basal diet supplemented with 0.5% fermented SBH (0.5% FSBH), 0.5% co-fermented SBH plus PESR (0.5% FSHP), and 1.0% co-fermented SBH plus PESR (1.0% FSHP) until 35 days of age. Compared with the control group, the 0.5% FSHP group showed significantly increased ratio of lactobacilli to *Clostridium perfringens* in the ceca, ileum villus height, and jejunum villus height-to-crypt depth ratio at 35 days of age. In conclusion, dietary supplementation of 0.5% FSHP in broiler chickens improved the body weight gain and intestinal morphology through the effects of its bioactive metabolites.

Recently it was found that suppression of IL-17F provides protection against colitis by inducing Treg cells through modification of the intestinal microbiota with increasing growth of *Lactobacillus murinus* by intereaction with low morecular β-glucan, such as laminaran and dectin 1 receptor [69].

#### 2.4.12. Application to the Skin

External application of APβG avoided muscle transplantation and promoted healing in cases with third-degree low-temperature burns [61]. APβG has been reported to serve cosmetic functions when applied to the skin. APβG has been reported to show anti-aging effects, whitening effects, and inhibitory effects on melanin production and tyrosinase activity.

Moreover, APβG reduced the activity of hyaluronidase, elastase, collagenase, MMP-1, and tyrosinase, as well as inhibited the production of melanin. Owing to its antioxidative and whitening effects, APβG may be used as a functional cosmetic material [49].

In UVB-induced hairless mice, APβG application to the skin reduced reactive oxygen species-mediated inflammation through its antioxidative and anti-inflammatory effects. APβG may be used as a raw material for cosmetics. Since antioxidants from natural sources may be an effective approach to prevent and treat UV-induced skin damage, the effects of purified APβG were evaluated in UVB-induced hairless mice. In that study, APβG containing 1.7% β-1,3/1,6-glucan, fibrous polysaccharides, and other organic materials, such as amino acids, and mono- and di-unsaturated fatty acids (linoleic and linolenic acids), showed anti-osteoporotic and immunomodulatory effects through antioxidative and anti-inflammatory mechanisms [54].

In traditional data, the effects of APβG on acute xylene-induced inflammation were observed. APβG at a dose of 62.5, 125, and 250 mg·kg^−1^ was orally administered once to xylene-treated mice (0.03 mL xylene was applied to the anterior surface of the right ear to induce inflammation). APβG showed a somewhat favorable effect in reducing the xylene-induced acute inflammatory response in mice [27].

#### 2.4.13. Osteoporosis Improvement

The anti-osteoporotic effects of APβG at doses of 31.25, 62.5, and 125 mg·kg^−1^ were investigated in ovariectomized mice. APβG was orally administered once daily for 28 days to bilateral ovariectomized mice, beginning four weeks after surgery. Changes in body weight, bone weight, bone mineral content, bone mineral density, failure load, histologic profiles, and histomorphometric parameters were evaluated, and serum osteocalcin, calcium, and phosphorus levels were measured. Alendronate was used as the reference drug. APβG significantly suppressed the decrease in bone weight, bone mineral content, failure load, bone mineral density, and serum calcium and phosphorus levels and increased serum osteocalcin levels in a dose-dependent manner. In addition, APβG significantly suppressed the decrease in histomorphometric parameters, such as volume, length, and thickness of the trabecular bone, the thickness of the cortical bone, and increased the number of osteoclasts in the femur and tibia. Although the effects of APβG were generally moderate and inferior to those of alendronate, its effects on the cortical bone thickness were superior. In addition, APβG exhibited favorable effects on ovariectomy-induced osteoporosis. However, further long-term studies are warranted to confirm the effects of APβG on osteoporosis [30].

Another study examined the synergistic anti-osteoporotic potential of mixtures containing different proportions of APβG, 13% β-glucan-containing extract, and *Textoria morbifera* Nakai (TM) extract compared with that of the single formulations of each herbal extract in bilateral ovariectomized mice, a well-known rodent model for studying human osteoporosis. The APβG:TM (3:1) formulation synergistically enhanced the anti-osteoporotic potential of single APβG or TM formulation, possibly due to the enhanced array of active ingredients. Furthermore, the effects of APβG:TM were comparable to those of RES (2.5 mg·kg^−1^). These results suggest that the APβG:TM (3:1) combination can serve as a novel pharmaceutical agent and/or functional food for managing osteoporosis in menopausal women [64].

The additional studies also assessed the beneficial skeletal muscle-preserving effects of extracellular polysaccharides from APβG on dexamethasone (DEXA)-induced catabolic muscle atrophy and osteoarthritis in mice [43,66].

#### 2.4.14. Use as a Material Inclusion Agent

APβG can solubilize pharmacologically active hydrophobic substances through complex formation without compromising their pharmacological activity. 1′-Acetoxychavicol acetate (ACA) is isolated from the roots of *Alpinia galanga* and shows various physiological activities such as antineoplastic and immunosuppressive properties. As a complex with APβGs, ACA can serve as a pharmaceutic agent, functional food material, and cosmetic material. The clathrate complex between ACA and APβG was successfully produced using a high-speed vibrational grinding method. The ACA–APβG complex was intraperitoneally administered to treat dinitrofluorobenzene-induced contact dermatitis in a mouse model, and auricle thickening, immune cell infiltration, and blood inflammatory cytokine levels were evaluated. Consequently, treatment with the ACA–APβG complex suppressed auricle thickening and immune cell infiltration and reduced the abundance of inflammatory cytokines in the blood, demonstrating anti-inflammatory effects [57].

Fullerenes are selected as target drugs, and their application in photodynamic therapy (PDT) is anticipated because of their high photosensitization ability; in this context, APβG was considered as a drug carrier. In addition, β-1,3-glucans are specifically recognized by dectin-1, which is a specific Toll-like receptor (TLR) expressed on macrophage surfaces. Therefore, the application of APβG is expected to increase the selectivity of PDT. Various fullerene derivatives were water-solubilized by APβG, and the efficacy of PDT using APβG as a drug carrier was evaluated in mouse peritoneal macrophages [62,67]. Upon complexing with APβG using a high-speed vibration pulverization method, fullerene, which is poorly water-soluble, could be efficiently solubilized. In atomic force microscopic images, the thickness of the APβG–C70 complex was about twice that of the APβG-only complex. The APβG–C70 complex functioned as an excellent photosensitizer both in vitro and in vivo and could specifically target macrophages expressing dectin-1 [62].

## 3. Discussion

### 3.1. Physiological Functions and Polysaccharides

As described earlier, β-1,3-glucan has traditionally been recognized as a functional food ingredient obtained from mushrooms and has been reported to possess antitumor activity. β-glucans are homopolysaccharides composed solely of glucose and have been reported to be functional in different binding modes. In addition to mushrooms, β-glucans have been derived from oat and barley, which contain repeating structures with β-1,3 and β-1,4 bonds in their primary chains without branching. Meanwhile, β-glucans derived from seaweeds, such as laminaran, contain repeating structures with β-1,3 and β-1,6 bonds. β-glucans with multiple linkages in their straight chains have been characterized to be highly water soluble [1,2,3,4,5].

As described above, the immunostimulatory mechanisms of β-glucans have recently been elucidated, and receptors on macrophages and dendritic cells, which are antigen-presenting cells, have been revealed to be involved in these mechanisms. The use of β-glucans as a functional food material to boost the intestinal immunity has been proposed, as discussed below. The mechanism underlying the immunostimulatory actions of APβG have also been clarified. Interestingly, the β-glucan polysaccharide, which is a macromolecule with a MW of a few million has been reported to show a triple helical structure. Recent studies have demonstrated that specific molecular weight is required for its action, and studies have accumulated the evidence of the association between APβG conformation and function, such as the recognition of β-1,6 bonds by antibodies, such as several kinds of IgG [31,33]. In addition, the higher-order structure of β-glucan has attracted much attention in host–guest chemistry based on its ability to solubilize poorly water-soluble hydrophobic substances, such as pharmaceutical compounds, vitamins, and physically or chemically treated nanoparticle materials. Therefore, the application of APβG as a drug delivery system is anticipated in vaccine adjuvant research.

### 3.2. Recognition Mechanism of β-Glucans

The immunomodulatory functions of β-glucans are realized through the interaction between the activation of innate immunity and the acquisition of adaptive immunity. Discoveries and evidence linking their roles are essential to elucidate the presence and function of receptors expressed on the surface of cells, such as macrophages, dendritic cells, and neutrophils. Among them, TLRs are specific receptors with some well-known ligands, such as nucleic acid molecules (DNA and RNA) and polysaccharides on the surface of flagella and cells. The specific receptors for β-glucan, in particular, include the β2-integrin-type complement receptor CR3 [70], lactosylceramide [71], and the C-type lectin dectin-1 [63], and there have been numerous recent studies focusing on these molecules. In the early 21st century, dectin-1 was recognized as a ligand for β-glucan. Using knockout mice, Brown et al. (2001) demonstrated that dectin-1, a C-type lectin, is expressed on the plasma membrane of mammalian leukocytes and is deeply involved in biodefense mechanisms, including phagocytosis, reactive oxygen species generation, and cytokine production; many studies on the roles of dectin-1 are currently underway [72]. Ohno et al. (2009) reported that different activation pathways have also been identified in the spleen [23,24,68]. The recognition structures include β-1,3-1,6-glucans with a β-1,3 primary chain with β-1,6 branches in fungi, linear β-1,3 and β-1,4-glucans in plants, and β-1,3-oligosaccharides with more than 16 sugars [4,73].

Anti-β-glucan IgG have been detected in vivo and reported to recognize β-1,6 branches [33]. Recently, the unique structure of APβG was determined, and it was revealed to be human serum antibody-reactive. The reactivity of APβG was stronger than that of *Grifola frondosa*-derived β-1,3-glucans but weaker than that of *Candida*-derived β-1,3-glucans. Specifically, APβG is reactive to human serum IgG antibodies, such as IgG2, IgG1, and IgG3.

APβG can bind to R-848, a low-molecular-weight ligand that activates TLR7. Moreover, APβG induced TNF-α and IL-12p40 and enhanced the phagocytotic ability of macrophages. In combination with R-848, β-glucan, which binds to the cell membrane-localized TLRs and the C-type lectin receptor dectin-1, could target THP-1 macrophages. Compared to R-848 treatment alone, co-treatment of R-848 and APβG significantly augmented TNF-α and IL-12p40 cytokine expression. These results suggest that APβG activates an array of innate immune receptor pathways to enhance the immune response of R-848 [47].

Stimulation with APβG effectively induced the interferon stimulated genes (ISGs) in macrophage-like cell lines, perhaps through the induction of IFNs and enhancement of STAT1-mediated transcriptional activation. The mRNA expression of ISGs was significantly increased in RAW264.7 cells following stimulation with APβG. APβG stimulation induced transient mRNA expression of IFN-β. Additionally, in IFN-α receptor-knockdown RAW264.7 cells, APβG stimulation induced the expression of viperin mRNA more efficiently than did IFN-α stimulation. Phosphorylation of Ser727, which is involved in promoting STAT1 activation, was rapidly increased following APβG stimulation. In addition, the expression of viperin mRNA induced by IFN-α stimulation was significantly enhanced by co-stimulation with APβG. These findings suggest that APβG stimulation effectively induces the expression of ISGs through the induction of IFNs and enhancement of STAT1-mediated transcriptional activation [48,50].

Based on these results, APβG has been reported to serve numerous health functions. Specifically, the antitumor and antimetastatic activities, antimicrobial effects, inflammation-reducing effects, food allergy-alleviating effects, and stress-relieving effects of APβG are known. Moreover, APβG improves lifestyle-related diseases by protecting the intestinal mucosa, reducing fat, lowering postprandial blood sugar, promoting bone and joint health, and alleviating gastric ulcers. APβG has been reported to induce HSP70. Interestingly, a randomized, single-blind, placebo-controlled, parallel-group clinical trial involving 65 participants (age, 22–62 years) with pollinosis was performed for three weeks from before and until the end of the cedar pollen season. In this trial, all participants consumed a bottle of placebo or APβG (150 mg) daily and recorded allergic symptoms in a diary. The APβG-administered group had a significantly reduced prevalence of sneezing, nose blowing, tears, and hindrance in the activities of daily life compared to the placebo group. These results suggest that APβG can serve as an effective treatment for allergic diseases [59].

## 4. Conclusions

The term “functional food” was coined in the 1980s in Japan [74]. A new and more widely established term used in Japan is “Food with Health Claims (FHC).” There are also a few definitions of functional foods. One of them is included in the final document of the research program Functional Food Science in Europe (FUFOSE), financed by the European Commission. According to this definition, food can be recognized as functional if its beneficial influence on one or more organism functions has been proven [75]. The need to continue the current research to systematize and verify the reports is warranted. This research should mainly focus on the practical application of β-glucan. There is still no precise information on the amount of β-glucan that should be consumed to obtain the desired health effect. Most observations were made using either very high doses of β-glucan or in combination with other biologically active ingredients. On this basis, it is difficult to unequivocally recommend the concentration of this ingredient that should be supplied daily to achieve the expected result. It is also unknown whether a potentially effective dose could be obtained by the consumption of products rich in β-glucan, or that the usage of the enrichment process of selected food products in this ingredient would be necessary. Therefore, consumers must be educated about β-glucan along with the benefits associated with their usual diet.

In the future, studies unveiling the precise mechanisms underlying the immunostimulatory functions of β-glucan as well as its effects on the autonomic nervous system and endocrine system are warranted. In addition, solubilized APβG is anticipated to serve as a functional food for modern and aging societies that experience high stress and as a potent pharmaceutical agent in the field of preventive medicine. Recently, owing to its unique triple helical structure, APβG has attracted much attention as a nanomaterial, hemostatic agent, and drug delivery system in the medical field.

## Figures and Tables

**Table 1 nutrients-13-00242-t001:** Some types of β-glucan with different structures derived from different natural sources.

Type of β-glucan Structure	Natural Source and Trivial Name of β-glucan
1,3-β-glucan (linear unbranched, homogeneous)	bacterium *Alcaligenes faecalis*, curdlan algae *Euglena gracilis,* paramylon fungus *Poria cocos*, pachyman grape *Vitis vinifera*, callose
1,3-1,6-β-glucan (linear with 1,6-linked β-glucosyl side branches)	algae *Laminaria* sp. laminarin (unbranched) algae *Eisenia bicyclis*, laminarin (some branched) fungus *Claviceps purpuria*, cell wall glucan fungus *Sclerotinia sclerotiorum*, cell wall or extracelluar glucan fungus (black yeast) *Aureobasidium pullulas*, extracellular glucan fungus/mushroom *schizophyllum commune*,extracelluar or cell wall glucan mushroom *Grifola frondosa*, cell wall glucan mushroom *Lentinula edodes*, cell wall glucan
1,3-1,6-β-glucan (branch on branch structure)	yeast *Saccharomyces cerevisiae*, cell wall glucan yeast *Candida albicans*, cell wall glucan yeast *Candia utilis*, cell wall glucan
1,3-1,4-β-glucan (linear)	cereal β-glucan, such as barley, oat, wheat, and rye Iceland moss *Centraria islandia*, lichenin
1,3-1,4-β-glucan (linear with 1,4-lincked β-glucosyl side branches)	oyster mushroom *Pleurotus ostreatus*, cell wall glucan

**Table 2 nutrients-13-00242-t002:** Various bioactive functions of β-1,3-1,6-glucan from black yeast *Aureobasidium pullulans* (APβG).

No.	Author, Year, Reference ^1^	Item of Chracterization or Healthy Function	Outline of Functionality and Efficacy
1	Arkadjeva 1969 [13]	Single cell protein	The first information on bioactive polysaccharide from *A. pullulans.*
2	Han et al., 1976 [14]	Intestinal immunity	Production of single-cell protein from cellulosic wastes using *A. pullulans. A. pullulans* cells were not toxic, and the values of their feed intake, weight gain, and protein efficiency ratio were superior to those of other cells.
3	T. Suzuki et al., 2004 [25]	Intestinal immunity	Intestinal immunostimulatory and modulatory effects. Cellular-level in vitro experiments using mouse lymphocytes from Peyer’s patch showed that immunoglobulin A was produced at APβG concentrations ranging from 0 to 200 μg/mL in a dose-dependent manner.
4	Kimura et al., 2006 [26]	Antitumor	Antitumor and antimetastatic actions in mice. The antitumor and antimetastatic actions of APβG may be involve in the enhancement of intestinal immune functions through the increases in NK- and IFN-gamma-positive cell numbers.
5	Kim et al., 2007 [27]	Reduction of inflamamatory	Reduction of the acute inflammatory responses induced by xylene application in mice. Xylene-induced acute inflammatory changes were significantly and dose-dependently decreased by beta-glucan treatment (up to 250 mg/kg).
6	Kimura et al., 2007 [28]	Anti food allergy	The anti-food allergic action of beta-glucan may be caused by the induction of IFN-g production in the small intestine and splenocytes. APβG diets (0.5–1%) significantly inhibited not only the OVA-specific IgE elevation but also reduced the production of IFN-g and the number of CD8- and IFN-gamma-positive cells from the splenocytes and in the small intestine, respectively.
7	Kimura et al., 2007 [29]	Anti stress	Inhibitory actions of APβG (100 mg/kg) on the increase in corticosterone level and reduction of NK activity induced by restraint stress. These effects may be associated with the abrogation of interleukin-6 (IL-6) and IL-12.
8	Shin et al., 2007 [30]	Reduction of osteoporosis	APβG exhibited favorable effects on ovariectomy-induced osteoporosis. It significantly and dose-dependently suppressed the decrease in bone weight, bone mineral content, failure load, bone mineral density, and serum calcium and phosphorus levels and the increase in serum osteocalcin levels.
9	Ikewaki et al., 2007 [31]	Immunomodulatory mechanism	APβG may have unique immune regulatory or enhancing properties. It stimulated the production of interleukin-8 (IL-8) or soluble Fas (sFas); however, it did not stimulate that of IL-1beta, IL-2, IL-6, IL-12 (p70 + 40), IFN-g, TNF-a, or soluble Fas ligand (sFasL), in either cultured PBMCs or cells of the human monocyte-like cell line U937.
10	Tada et al., 2008 [20]	Structure information	The primary structure of APβG and its biological activities were determined and evaluated, respectively using NMR spectroscopy. The structure comprises a mixture of a 1-3-beta-d-glucan backbone with single 1-6-beta-d-glucopyranosyl side-branching units in every two residues (major structure) and a 1-3-beta-d-glucan backbone with single 1-6-beta-d-glucopyranosyl side-branching units in every three residues (minor structure).
11	Tada et al., 2009 [32]	Immunomodulation mechanism	Immunomodulatory effect of APβG on DBA/2 mouse-derived splenocytes *in vitro*. APβG strongly induced the production of various cytokines, especially Th1 cytokines (e.g., IFN-g and IL-12p70) and Th17 cytokines (e.g., IL-17A), but did not induce the production of IL-4, IL-10, and TNF-a on splenocytes in vitro.
12	Tada et al., 2009 [33]	Immunomodulatory mechanism	This is the first study in which the branched chains at position 6 of beta-D-glucan strongly contribute to its recognition by antibodies in human sera. APβG reacted to IgG antibodies in human sera and the IgGs recognize branched chains at position 6.
13	Kimura et al., 2009 [34]	Protection of intestine	Protective effects of APβG (50 and 100 mg/kg twice daily) against the toxicity of UFT (combination of tegafur (1-(2-tetrahydrofuryl)-5-fluorouracil) and uracil) in mice bearing colon 26 tumors. Histological analysis showed that the damage found in the villi of the small-intestine by UFT was inhibited by the orally administered beta-glucan.
14	Sumiyoshi et al., 2009 [35]	Reduction and control of blood glucose	Reduction and control of blood glucose level in mice. In the 100 mg/kg and 200 mg/kg APβG dose groups, the increase in blood glucose from 15 to 30 min after glucose administration was minimal, after which the blood glucose level decreased, and significantly decreased 60 min after administration.
15	Yoon et al., 2010 [36]	Immunomodulatory mechanism	Immunomodulatory effects of exopolymers of *A. pullulans* containing APβG, which were orally administered at 10 mL/kg, were evaluated on cyclophosphamide (CPA)-treated mice. APβG can be effectively used to prevent an immunosuppress mediated (at least partially) and the recruitment of T cells and TNF-a-positive cells or enhancement of their activity.
16	Tada et al., 2011 [37]	Immunomodulatory mechanism	Immunomodulatory production of various cytokines in DBA/2 mouse-derived splenocytes in vitro was found via dectin-1-independent pathways. The production of IFN-γ in DBA/2 mouse-derived splenocytes by APβG was not inhibited following a treatment with an anti-dectin-1 neutralizing antibody.
17	Tada et al., 2011 [38]	Immunomodulatory mechanism	The induction of cytokines by APβG was dependent on the existence of a granulocyte macrophage colony-stimulating factor (GM-CSF). GM-CSF is indispensable for the induction of cytokines by APβG in mouse-derived splenocytes, similar to a typical 1,3-β-d-glucan from *Sparassis crispa* (SCG).
18	Tanaka et al., 2011 [39]	Reduction of ulcer	Oral administration of APβG (>100 mg/kg) ameliorated the gastric lesions induced by ethanol (EtOH) or HCl in mice. The administration of APβG also suppressed EtOH-induced inflammatory responses through the protection of the gastric mucosa from the formation of irritant-induced lesions by increasing the levels of defensive factors, such as HSP70 and mucin.
19	Muramatsu et al., 2012 [40]	Anti virus	Oral administration of *A. pullulans*-cultured fluid enriched with APβG exhibits efficacy in protecting mice infected with a lethal titer of the A/Puerto Rico/8/34 (PR8; H1N1) strain of the influenza virus.
20	Ku et al., 2012 [41]	Anti allergy	The effect of APβG, orally administered at 125 mg/kg, on ovalbumin (OVA)-induced allergic asthma was found in OVA-inducing asthmatic mice. The increase in body weight after OVA aerosol challenge, lung weight, total leukocytes and eosinophils in peripheral blood, total cell numbers, and neutrophil and eosinophils in BALF were detected in the OVA control compared to sham control (non-OVA).
21	Uchiyama et al., 2012 [42]	Microbiome	The effects of oral administration on bacterial flora in the intestines of domestic animals, using Holstein cows and newborn Japanese Black calves, were observed. The expressions of TNF-α and interleukin (IL)-6 in all cows became slightly lower and the bacterial flora were tendentiously changed.
22	Kim et al., 2012 [43]	Reduction of osteoarthritis	Osteoarthritis (OA) was effectively induced by anterior cruciate ligament transection and partial medial meniscectomy (ACLT&PMM) by APβG (42.5mg/kg).
23	Sato et al., 2012 [44]	Anti allergy	Effective therapeutic treatment of allergic diseases, inhibition of mast cell degranulation, and passive cutaneous anaphylaxis (PCA) were shown. APβG (100 to 150 mg/kg) dose-dependently inhibited the degranulation of both rat basophilic leukemia (RBL-2H3) and cultured mast cells (CMCs) activated by calcium ionophore A23187 or IgE.
24	Tanioka et al., 2012 [45]	Anti microorganism	Positive effect of oral administration of APβG on *Candida albicans* or methicillin-resistant *Staphylococcus aureus* (MRSA) infection in immunosuppressed mice fed 2.5% APβG diet for 14–30 days.
25	Tanioka et al., 2013 [46]	Immunomodulatory mechanism	APβG had effects on intestinal immune systems by Peyer’s patch (PP) cells and interleukin (IL)-5, IL-6, and IgA production in culture media. The production of IL-6 and IgA by PP cells and that of IL-6 by PP dendritic cells (PPDCs) in APβG-fed and cyclophosphamide (CY)-treated mice also increased.
26	Tamegai et al., 2013 [47]	Immunomodulatory mechanism	Activation of several distinct innate immune receptor signaling pathways enhances the immune response induced by R-848, indicating non-influenza antiviral efficacy. The expression of TNF-a and IL-12p40 was significantly enhanced when co-stimulated with culture supernatants of R-848 and APβG compared with the culture supernatant of R-848 alone.
27	Iwai et al., 2013 [48]	Anti virus	Antiviral effects of the expression of interferon-inducible genes, through the induction of interferon, and the enhancement of the transcriptional activity of STAT1 were observed.
28	Kim et al., 2014 [49]	Anti oxidant	Several antioxidants may serve as a functional ingredient in cosmetic products by reducing hyaluronidase, elastase, collagenase, and MMP-1 activities and inhibiting melanin production and tyrosinase activities.
29	Muramatsu et al., 2014 [50]	Immunomodulatory mechanism	Stimulation with APβG effectively induces the interferon (IFN) stimulated genes (ISGs) in macrophage-like cell lines through the induction of IFN and the enhancement of STAT1-mediated transcriptional activation.
30	Ganesh et al., 2014 [51]	Reduction of triglyceride and cholesterol, in human	In a clinical study, one male was orally administered 1.5 mg of APβG for two months, and his triglyceride, VLDL, and HDL cholesterol levels decreased from 523 mg/dL to 175 mg/dL, 104.6 mg/dL to 35 mg/dL, and 27 mg/dL to 38 mg/dL, respectively.
31	Oboshi et al., 2014 [52]	Immunomodulatory mechanism	When AGβP-containing foods were orally administered to mice (BALB/c six-week old females), an increase in the titer of antibodies in the blood and the phagocytic capacity of blood macrophages were observed.
32	Aoki et al., 2015 [53]	Reduction of triglyceride and cholesterol	Oral administration of APβG is effective in preventing the development of high-fat diet (HFD)-induced fatty liver in mice. After 16 weeks of oral administration of APβG, serological analysis showed that HFD-induced high blood cholesterol and triglyceride levels were reduced by the oral administration of APβG. HFD induced-fatty liver was also significantly reduced.
33	Kim et al., 2015 [54]	Reduction osteoporosis	The effects of purified exopolymers from APβG were evaluated in UVB-induced hairless mice. E-AP-SM2001 consists of 1.7% β-1,3/1,6-glucan, fibrous polysaccharides, and other organic materials, such as amino acids and mono- and di-unsaturated fatty acids (linoleic and linolenic acids), and shows anti-osteoporotic and immunomodulatory effects through antioxidant and anti-inflammatory mechanisms.
34	Kawata et al., 2015 [55]	Anti-tumor	Mechanism of anti-tumor activities in mice have been demonstrated. Stimulation with APβG induces TRAIL expression in mouse and human macrophage-like cell lines. TRAIL is known to be a cytokine that specifically induces apoptosis in transformed cells, but not in untransformed cells.
35	Lai et al., 2015 [56]	Improvement of intestine morphology	The effects of co-fermented *Pleurotus eryngii* stalk residues (PESR) and soybean hulls with *A. pullulans* on performance and intestinal morphology of broiler chickens significantly increased the ratio of lactic acid bacteria to *Clostridium perfringens* in ceca, ileum villus height, and jejunum villus height/crypt depth ratio of 35-day old birds.
36	Li et al., 2015 [57]	Anti inflammatory effect	Anti-inflammatory effect of water solubilized 1′-Acetoxychavicol acetate (ACA) on contact dermatitis by complexation with β-1,3-glucan isolated form *A. pullulans* black yeast was reported.
37	Aoki et al., 2015 [58]	Reduction of atherosclerosis	The effect of oral administration of APβG on high-fat diet (HFD)-induced atherosclerosis was evaluated using apolipoprotein E deficient mice, a widely used mouse model for atherosclerosis. HFD-induced atherosclerosis was significantly reduced in the APβG-treated mice.
38	Jippo et al., 2015 [59]	Anti allergy in human	A clinical study, comprising a randomized, single blind, placebo-controlled, and parallel group, was performed in 65 subjects (aged 22 to 62). All subjects consumed one bottle of placebo or beta glucan (150 mg) daily and their allergic symptoms were recorded in a diary. APβG group had a significantly lower prevalence of sneezing, nose blowing, tears, and hindrance to the activities of daily living.
39	Iinuma et al., 2016 [60]	Anti colongenomic cancer	APβG was orally administered in combination with anticancer drugs, such as Avastin, Elplat, Levofolinate and fluorouracil, to treat for stage III colon cancer patients, on which two sites of cologenomic cancer were transferred, and remarkable cancer elimination was reported in the all cases.
40	Hirabayashi et al., 2016 [21]	Structural characterization of by hydrothermal treatment	The chemical structure of hydrothermally treated APβG was characterized using techniques such as gas chromatography/mass spectrometry (GC/MS) and nuclear magnetic resonance (NMR). It became water soluble, with an average molecular weight of 128,000 and was completely hydrolyzed to glucose by enzymatic reaction. Gentiobiose and glucose were released as products during the reaction with the maximum yield approximately 70% (*w*/*w*) of gentiobiose.
41	Yamamoto et al., 2017 [61]	Wound healimg	Muscle transplantation was avoided, and rapid healing was observed when there was external application of APβG in the case of third degree deep low temperature burn.
42	Ikeda et al., 2017 [62]	Inclusion of [63] Fullerene	[63] Fullerene was dissolved in water via complexation with β-1,3-glucan using a mechanochemical highspeed vibration milling apparatus. The photodynamic activity of APβG-complexed C70 was highly dependent on the expression level of dectin-1 on the cell surfaces of macrophages. The photodynamic activity increased owing to the synergistic effect between β-1,3-glucan-complexed 1′-acetoxychavicol acetate and the C70 complex.
43	Cho et al., 2018 [64]	Anti osteoporotic	The synergistic anti-osteoporotic potential of mixtures containing different proportions of APβG and TM compared with that of single formulations of each herbal extract using bilateral ovariectomized (OVX) mice. The EAP:TM (3:1) formulation synergistically enhanced the anti-osteoporotic potential of individual EAP or TM formulations, possibly owing to the enhanced variety of active ingredients.
44	Fujikura et al., 2018 [65]	Anti virus	APβG exhibits adjuvant activity and renders mice to be resistant to influenza A virus infection. Intraperitoneal administration of APβG increased the serum level of IL-18 and the number of splenic IFN-γ producing CD4+ cells during influenza A virus infection. The adjuvant effect of APβG was distinct from that of alum. In addition, AP-CF injection barely increased the number of peritoneal neutrophils and inflammatory macrophages.
45	Lim et al., 2018 [66]	Muscle preserving effect	The beneficial skeletal muscle‑preserving effects of extracellular polysaccharides from APβG on dexamethasone (DEXA)‑induced catabolic muscle atrophy in mice. APβG at 400 mg/kg exhibited favorable muscle protective effects against DEXA‑induced catabolic muscle atrophy; the effects are comparable with those of oxymetholone (50 mg/kg).
46	Hayashi et al., 2019 [22]	Anti inflammatory effect	Inflammatory immunostimulatory action of nitrogen oxide (NO) synthesis and TNF a production in RAW264.7 cells induced by Lipopolysaccharide (LPS) were suppressed by APβG with lower molecular weight less than 10,000.
47	Hino et al., 2019 [67]	Inclusion of porphyrin	Fluorescence intensities of water-soluble APβG-complexed porphyrin derivatives were very weak owing to self-quenching. However, APβG-complexed tetra (aminophenyl) porphyrin exhibited ‘off-state’ to ‘on-state’ fluorescence switching activity via intracellular uptake. Furthermore, the internalized complex showed a high level of photodynamic activity toward HeLa cells under photoirradiation at long wavelengths.

^1^ Reference numbers in “Author, Year and Reference” column is shown in reference numbers in the text.

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
