# Peer review of "Biological Activity of High-Purity β-1,3-1,6-Glucan Derived from the Black Yeast Aureobasidium pullulans: A Literature Review"

_nutrients, 2021, doi:10.3390/nu13010242_

Round 1

Reviewer 1 Report

The authors summarize and comment the recent findings on the effect of beta-1,3-1,6 glucan derived from Aureobasidium pullulans on physiological functions of the organism. To my knowledge, based on the Pubmed search there are no review papers on this or similar topic, which makes the article not only relevant to the special issue "Nutrition and Metabolism" but possibly highly recognized by other authors interested in the biological effects of pullulan. However, there are several aspects that needs explanation such as:

  1. Please consider changing a title to Biological activity of high-purity β-1,3-1,6-2glucan derived from the blackyeast Aureobasidium pullulans: A literature review.
  2. Please provide the inclusion/exclusion criteria for articles in section Materials/Methods.
  3. The authors refer to authorities from Japan ("Survey and Study on Review of Safety of Existing Food Additives" by the Japan Food Chemistry Research Foundation of the Ministry of Health, Labour and Welfare Foundation). Can you please add information of the status of black yeast products in Europe, US.
  4. Please define "various cytokines" in line 193. Their nature is important to estimate, if the effect of the b-glucan facilitate the pro- or anti-inflammatory responses.
  5. Headline in line 116 (3.1.2β-. glucan) needs minor editing.
  6. I would suggest renaming the section 3.4.4. to Antimicrobial activity and introducing MRSA into text.
  7. Please consider adding some introduction to section 3.4.6 that would occur before "In a clinical study, a male subject was orally administered 1.5mg APbG for 2 months." or consider placing this sentence after next one which introduces a Reader to this section.
  8. In section 3.4.9. there is one citation. Please explain, if the whole section is desribing one study. Are there any other data on this biological function of the APbG?
  9. In line 447 please correct the latin name for Clostridium perfringens.
  10. Please add italics to Alpinia galanga in line 510.
  11. In my opinion, the information included in discussion: 4.2 should be combined within the findings in sections 3.49 or 3.42, and paragraph 4.3 should be combined with 3.4.1.

Author Response

Reviewer 1

Dear Reviewer 1,

Thank you for your comments and indications. We revised our article alomg with yours  considering another reviewer 2 comments. Our revised portions are highlighted in our revised manuscript in yellow color and please check them. Detail points are described below;

I would like to ask you review our revised manuscript once again.

Thank you for your attention.

With best regards,

Toshio SUZUKI

As a corresponding author

  1. Please consider changing a title to Biological activity of high-purity β-1,3-1,6-2glucan derived from the black yeast Aureobasidium pullulans: A literature review.

--> We revised as your indication in Title.

  1. Please provide the inclusion/exclusion criteria for articles in section Materials/Methods.

--> Materials/Methods was delete and combined in Introduction session as searching method information (in line 109-115). And results are added in Table 2, which was as supplement data in the previous submit, as summarized data with more detailed information “Item of Chracterization or healthy function” in line 212.

  1. The authors refer to authorities from Japan ("Survey and Study on Review of Safety of Existing Food Additives" by the Japan Food Chemistry Research Foundation of the Ministry of Health, Labour and Welfare Foundation). Can you please add information of the status of black yeast products in Europe, US.

--> Beta glucan from Aureobasidium pillulas was mainly developed in Japan around latter 1980 so that there are a lot of information in Japan. On the other hand, this fungus is famous for a resource of Pullula with alpha-1,4-1,6-glucan. Therefore, In US and EU, A. pullulans is admitted Pullulan from A. pullulans. This information is added in line 198-200.

  1. Please define "various cytokines" in line 193. Their nature is important to estimate, if the effect of the b-glucan facilitate the pro- or anti-inflammatory responses.

--> I checked the corresponding reference again and confirm that IgA and, IL5 andIL6 in vivo and vitro was enhanced respectively. These information are added in lines 240-241 and 243-245.

  1. Headline in line 116 (3.1.2β-. glucan) needs minor editing.

-> We revise as “Beta-glucan recovery and purification” in line 135.

  1. I would suggest renaming the section 3.4.4. to Antimicrobial activity and introducing MRSA into text.

--> We revised as your indication in line 300-302.

  1. Please consider adding some introduction to section 3.4.6 that would occur before "In a clinical study, a male subject was orally administered 1.5mg APbG for 2 months." or consider placing this sentence after next one which introduces a Reader to this section.

-> We revised as follows;

1) The section title is changed to “Effects on improving lifestyle-related diseases, including obesity” to clear information in this section..

2) Data and contents (paragraph) are changed from efficacies with animals to with human in order in lene 338-356.

3) Anti obesity mechanism using beta-glucan from A. pullulans are added in line 345-348.

4) We added information of mode of beta-glucan from oats and barley for anti obesity in line 59-65.

  1. In section 3.4.9. there is one citation. Please explain, if the whole section is desribing one study. Are there any other data on this biological function of the APbG?

---> Anti stress effect and its relations to immunity is cited in this reference. And then, further clinical data along with this result were conducted. The details in clinical data, which is unpublished in original article are add in line 459-466.

  1. In line 447 please correct the latin name for Clostridium perfringens.

-> We revised as your indication in line 510.

  1. Please add italics toAlpinia galanga in line 510.

--> We revised as your indication in line 576.

  1. In my opinion, the information included in discussion: 2 should be combined within the findings in sections 3.49 or 3.42, and paragraph 4.3 should be combined with 3.4.1.

---> We revised as follows along wth your opinion;

  • The section “2 related intestinal immunity” was combined with section “2.4.1. Intestinal immunostimulatory effect” as introduction about intestinal immunity in line 217-235.
  • About section “3. Recognition Mechanism of β-glucans, we remained as summarized efficacies and the latest recognition mechanism of beta-glucan from A. pullulans in Discussion session.

Reviewer 2 Report

I must say that the article deals with a significant biotechnological problem. The authors undertook to explain and characterize the possibility of obtaining β-glucan from Aureobasidium pullulans.

β-glucan is a polysaccharide belonging to the soluble fraction of dietary fiber. Due to its documented beneficial effects on health, it is considered a functional food ingredient. The most important benefits for the human body associated with the consumption of this ingredient include lowering cholesterol, achieved mainly due to the ability of β-glucan to: bind bile acids, stimulate immunomodulatory cytokines, regulate the level of interleukin 1β (IL-1β) and the cytochrome P450 isoenzyme CYP7A1 (CYP7A1) and reducing cholesterol absorption. Other benefits include: lowering blood sugar levels, supporting immunity, preventing the development of obesity or regulating intestinal function. However, there is a need for further research that would confirm the currently observed properties of β-glucan and perhaps demonstrate new benefits resulting from its consumption.

The article is interesting and well written. In my opinion, however, the authors should compare the obtaining of β-glucan with the association of other yeasts (eg Saccharomyces, Candida). This information will fill the topic more fully.

The authors should describe in the manuscript that in 1999 Functional Food Science in Europe (FUFOSE) gave a complete, albeit working definition of functional food, according to which such food above the nutritional effect must have a beneficial effect on improving health, well-being and health, proven by representative scientific studies. / or reducing disease risk. This applies to β-glucan and its uses.

The authors did not mention that the biological activity of β-glucan depends on its: molecular weight, size, frequency of connections, changes in structure, conformation and solubility.

It is still unclear how β-glucan can have a beneficial effect on the lipid balance. There are several hypotheses for the hypocholesterolemic activity of β-glucan, each of which appears to be scientifically valid. The authors should develop and describe this idea

Add these publications to your introduction describing the possible preparation of β-glucan. This will strengthen the work and interest the readers:
Bzducha-Wróbel, A., Pobiega, K., Błażejak, S., & Kieliszek, M. (2018). The scale-up cultivation of Candida utilis in waste potato juice water with glycerol affects biomass and β (1, 3) / (1, 6) -glucan characteristic and yield. Applied microbiology and biotechnology, 102 (21), 9131-9145.
Bzducha-Wróbel, A., Koczoń, P., Błażejak, S., Kozera, J., & Kieliszek, M. (2020). Valorization of deproteinated potato juice water into β-glucan preparation of C. utilis origin: comparative study of preparations obtained by two isolation methods. Waste and Biomass Valorization, 11 (7), 3257-3271.

It is worth adding information about the possible industrial production of this component of the cell wall.

2. Materials and Methods - I would delete this section and prepare the article as an overview. Modify individual subsections.

3.4.4. Antibacterial (Methicillin-resistant Staphylococcus aureus, MRSA) and antifungal (Candida sp.) Activities - the names of microorganisms are written in italics. Check the full article.

3.4.11. Effects on the microbiome - this chapter is very poorly written. The latest literature data is missing. The authors should elaborate on this.

No prospects for the future. Authors should write this more paying attention to the latest literature data.

In conclusion, the authors should describe, for example, such a summary: There is a need to continue the research currently being conducted, which could systematize and verify the current observations. They should go mainly towards the practical application of β-glucan. There is still no precise information on the amount of β-glucan that should be consumed to obtain the desired health effect. The vast majority of observations were made was either using very high doses of β-glucan or in combination with another biologically active ingredient. On their basis, it is difficult to unequivocally recommend to the consumer how much of this ingredient should be supplied daily to achieve expected result. It is also unknown if a potentially effective dose could be obtained as a result of the consumption of products rich in β-glucan, or it will be necessary to use the enrichment process of selected food products in this ingredient. Therefore, a priority action should be to educate consumers about β-glucan itself and the benefits associated with its introduction go to your usual diet.

Add doi numbers to references.

Author Response

Reviewer 2

Dear Reviewer 2,

Thank you for your comments and indications. We revised our article alomg with yours  considering another reviewer 1 comments. Our revised portions are highlighted in our revised manuscript in yellow color and please check them. Detail points are described below;

I would like to ask you review our revised manuscript once again.

Thank you for your attention.

With best regards,

Toshio SUZUKI

As a corresponding author

  1. The article is interesting and well written. In my opinion, however, the authors should compare the obtaining of β-glucan with the association of other yeasts (eg Saccharomyces, Candida). This information will fill the topic more fully.

-> The information for yeast beta-glucan was add in Introduction session in line 66-76, and then, Table 1 as summarized information of resources of several kinds of beta-glucan including that of yeast in line 50. There are some comparison between beta glucan from yeast and black yeast such as structure, solubility and so on.

  1. The authors should describe in the manuscript that in 1999 Functional Food Science in Europe (FUFOSE) gave a complete, albeit working definition of functional food, according to which such food above the nutritional effect must have a beneficial effect on improving health, well-being and health, proven by representative scientific studies. / or reducing disease risk. This applies to β-glucan and its uses.

--> The general and definition information of “Functional Foods” in Japan and EU is added in line 673-678.

  1. The authors did not mention that the biological activity of β-glucan depends on its: molecular weight, size, frequency of connections, changes in structure, conformation and solubility.

--> About biological activities depending on several physical-chemical characterization, the information of biological activity on frequency of connection and molecular size are added in line 77-83 and line 182-189, respectively.

  1. It is still unclear how β-glucan can have a beneficial effect on the lipid balance. There are several hypotheses for the hypocholesterolemic activity of β-glucan, each of which appears to be scientifically valid. The authors should develop and describe this idea

--> The beneficial effect of beta-glucan on lipid balance are summarized and added as follows;

1) On information of beta glucan from oat and barley, we show in line 59-65.

2) On information of beta glucan from A. pullulans, we show in line 344-348.

  1. Add these publications to your introduction describing the possible preparation of β-glucan. This will strengthen the work and interest the readers:
    Bzducha-Wróbel, A., Pobiega, K., Błażejak, S., & Kieliszek, M. (2018). The scale-up cultivation of Candida utilis in waste potato juice water with glycerol affects biomass and β (1, 3) / (1, 6) -glucan characteristic and yield. Applied microbiology and biotechnology, 102 (21), 9131-9145.
    Bzducha-Wróbel, A., Koczoń, P., Błażejak, S., Kozera, J., & Kieliszek, M. (2020). Valorization of deproteinated potato juice water into β-glucan preparation of C. utilis origin: comparative study of preparations obtained by two isolation methods. Waste and Biomass Valorization, 11 (7), 3257-3271.

It is worth adding information about the possible industrial production of this component of the cell wall.

----> We revised as your indication in Table 1 (line 50) and line 75-76.

  1. Materials and Methods - I would delete this section and prepare the article as an overview. Modify individual subsections.

----> We delete session “Materials/Methods” and combine in “Introduction” session. And then, overview information about beta glucan from cereals and yeast (line 59-83), Table 1 (line 50) and Table 2 as summary of “Outline of functionality and efficacy” (line 212) are added.

  1. 4.4. Antibacterial (Methicillin-resistant Staphylococcus aureus, MRSA) and antifungal (Candida sp.) Activities - the names of microorganisms are written in italics. Check the full article.

--> We revised as your indication in line 300-309.

  1. 4.11. Effects on the microbiome - this chapter is very poorly written. The latest literature data is missing. The authors should elaborate on this.

No prospects for the future. Authors should write this more paying attention to the latest literature data.

--> On influence for microbiota by beta glucan, the latest mechanisms of cereal beta glucan with beta 1,3-1,4-glucan and laminaran with beta 1,3-1,6-glucan are added in line 59-65 and line 522-524, respectively.

  1. In conclusion, the authors should describe, for example, such a summary: There is a need to continue the research currently being conducted, which could systematize and verify the current observations. They should go mainly towards the practical application of β-glucan. There is still no precise information on the amount of β-glucan that should be consumed to obtain the desired health effect. The vast majority of observations were made was either using very high doses of β-glucan or in combination with another biologically active ingredient. On their basis, it is difficult to unequivocally recommend to the consumer how much of this ingredient should be supplied daily to achieve expected result. It is also unknown if a potentially effective dose could be obtained as a result of the consumption of products rich in β-glucan, or it will be necessary to use the enrichment process of selected food products in this ingredient. Therefore, a priority action should be to educate consumers about β-glucan itself and the benefits associated with its introduction go to your usual diet.

-----> We revised as your indication in Discussion session in 678-687. 

  1. Add doi numbers to references.

----> We revised as your indication in reference as much as possible.

Round 2

Reviewer 2 Report

The manuscript has been revised. However, be careful that the authors should adapt the article to the new periodicals.

The article should be adapted to the new journal template.
https://www.mdpi.com/journal/nutrients/instructions